# HISTOPATHOLOGY LEARNING DATASET AUGMENTATION: A HYBRID APPROACH

**Ramzi HAMDI**[1]                                             RAMZI.HAMDI@WITSEE.AI

**Clément BOUVIER**[1]                                   CLEMENT.BOUVIER@WITSEE.AI

**Thierry DELZESCAUX**[2]                            THIERRY.DELZESCAUX@CEA.FR

**Cédric CLOUCHOUX**[1]                             CEDRIC.CLOUCHOUX@WITSEE.AI

[1] *WITSEE, Neoxia, Paris, France*

[2] *CEA-CNRS-UMR 9199, LMN, MIRCen, Univ. Paris-Saclay, Fontenay-aux-Roses, France*

**Editors:** Under Review for MIDL 2021

## Abstract

In histopathology, staining quantification is a mandatory step to unveil and characterize disease progression and assess drug efficiency in preclinical and clinical settings. Supervised Machine Learning (SML) algorithms allow the automation of such tasks but rely on large learning datasets, which are not easily available for pre-clinical settings. Such databases can be extended using traditional Data Augmentation methods, although generated images diversity is heavily dependent on hyperparameter tuning. Generative Adversarial Networks (GAN) represent a potential efficient way to synthesize images with a parameter-independent range of staining distribution. Unfortunately, generalization of such approaches is jeopardized by the low quantity of publicly available datasets. To leverage this issue, we propose a hybrid approach, mixing traditional data augmentation and GAN to produce partially or completely synthetic learning datasets for segmentation application. The augmented datasets are validated by a two-fold cross-validation analysis using U-Net as a SML method and F-Score as a quantitative criterion.

**Keywords:** Generative Adversarial Networks, Histology, Machine Learning, Whole Slide Imaging

## 1. Introduction

Histology is widely used to detect biological objects with specific staining such as protein deposits, blood vessels or cells. Staining quantification heavily relies on manual techniques, a tedious and time-consuming task. Therefore, automated methods, among which supervised machine learning SML (1), are increasingly used to automatically detect and quantify biological structures. One of the biggest issues facing the use of SML in medical imaging is the lack of large, labelled and available datasets. Not only manual annotation is time-consuming, but it is also highly dependent on the bias of the experts. The limited amount of annotated data can decrease the performance of SML algorithms which often need large quantities of learning data to avoid overfitting.

To overcome these issues, learning datasets can be artificially increased using data augmentation (2). A number of techniques have been proposed in the last decade, based on traditional methods such as Rotation (3), RandomGaussBlur (4), RandomAffine (5), Elastic Distortion (6) or HEDJitter (7). Although opening the way to using synthetic data to compensate for limited dataset availability, the mentioned methods suffer from several

problems, among which non-genericity: most of the proposed methods were designed and optimized for specific datasets and markers. Another issue is related to their intrinsic nature: standard data augmentation produces only limited alternative data, depending on the hyperparameters intervals and degrees of freedom. As a result, transformed images have a similar distribution or spatial diversity than the original ones, leading to limited performance improvement. However Generative Adversarial Networks (GAN) offer a new way to generate synthetic samples to extend and diversify datasets. The original method was based on the competition of two networks: a Generator network aimed at producing realistic images to fool a Discriminator network (8). Deep Convolutional GAN (DC-GAN) (9) consisted in convolution layers without max pooling or fully connected layers.

The presented work aimed at mixing traditional Data Augmentation and GAN methods to ensure sufficient databases generation for high-quality histopathological staining segmentation. The segmentation was performed with U-Net (10) and the model was validated using two-fold cross-validation.

## 2. Material and Methods

### 2.1. Dataset and computational environment

#### 2.1.1. Histological dataset

This study was performed accordingly with the principles of the Declaration of Helsinki. Approval was granted by the CETEA (Comité d'éthique en expérimentation animale) n°44 and the Ministry of higher education, research and innovation (MESRI).

Dataset was composed of 114 histological sections extracted from a 13.5-months-old mouse amyloidosis model (APP/PS1dE9) brain, stained with beta Amyloid Monoclonal Antibody (BAM-10) and counter-stained with Bluing Reagent (11). From the digitized sections, 1600 images of 128x128 pixels each were randomly extracted. An expert segmented them in two classes: 1- background and Bluing Reagent stained tissue and 2- BAM-10 stained tissue.

#### 2.1.2. Learning and Test datasets

The dataset described in section 2.1.1 was split in half in a learning (800 couples, image/segmentation) and a test (800 couples) datasets (Figure.1-a). Each initial learning dataset used was built by randomly selecting 90 from the learning dataset.

#### 2.1.3. Computational environment

Algorithms were run on the following hardware setup: 128 GB RAM, i9-7980XE CPU, Nvidia Quadro P5000 graphics card. Software environment included 450.51.06 nvidia driver, cuda version 10.0. hosted on an UBUNTU 16.04 operating system. Algorithms were coded in python and implemented on Keras framework.

### 2.2. Data augmentation

#### 2.2.1. Traditional Data Augmentation methods

The first method was based on rotation, reversal (3) and a custom-made circular translation method. The latter consisted in splitting an image into 2 parts, blending one part with the

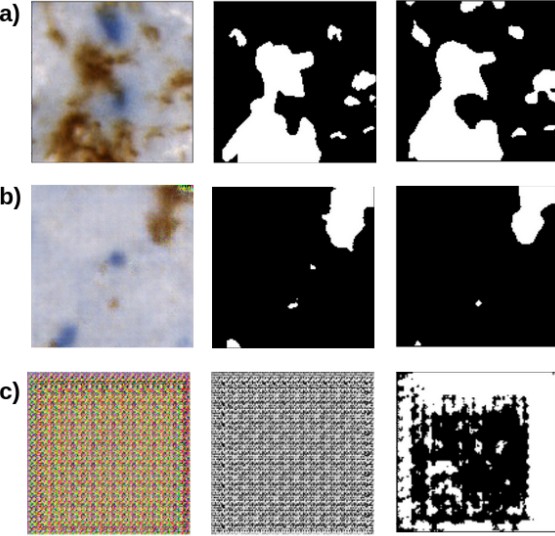

Figure 1: Example images and corresponding masks. From left to right: a) Initial dataset: raw data, manual segmentation, U-Net segmentation. b) DC-GAN trained with an intermediate dataset: generated image, generated mask, U-Net segmentation. c) DC-GAN trained with the initial dataset: generated image, generated mask, U-Net segmentation.

reversal of the other using the Gaussian Laplacian Pyramid Blend algorithm (12). The other used methods were Random Affine (4), Random Gauss Blur (5), Random Elastic transformation (6) and HEDJitter randomly disturbed HED color space values (7). During the entire process, these traditional methods generated 149,600 couples of images with their corresponding segmentation in two-fold cross-validation (13) (29,920 with each method). Structural similarity index measure (SSIM) is a value scoring the spatial similarity between two images. This index, not rotation invariant (14), ranges from 0 (images not similar) to 1 (identical images). The SSIM was computed 5 times between the initial learning dataset and intermediate method for each traditional method. Then each method was scored by the average SSIM and its standard deviation. Average SSIM computation was performed in a two-fold cross-validation setting.

### 2.2.2. DC-GAN Data Augmentation

The GAN Data Augmentation was performed using the DC-GAN algorithm (9). The quantity of the DC-GAN generated images was equal to the quantity of the generated images using traditional methods in the intermediate dataset (between 880 and 3600 generated images depending on the protocol, detailed section 2.2.3). Image generation started after the 360th epochs and 100 images per epoch, until the generation of the final dataset. These numbers were chosen empirically after a testing phase and interpretation of the epoch and the number of images per epoch. Generated raw images were smoothed using a standard

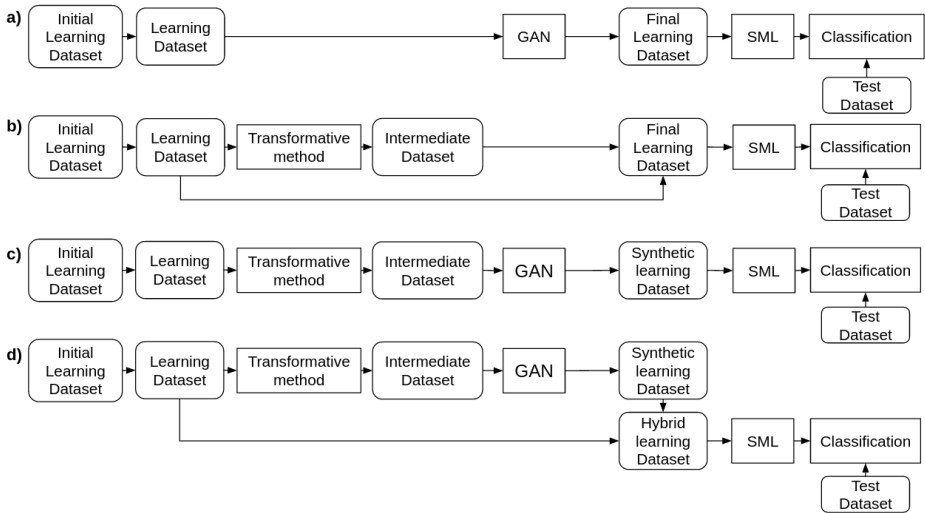

Figure 2: Scheme of the 4 learning protocols. a) GAN only. b) Traditional method only. c, d) Mix between Traditional method and GAN

median filtering, with a kernel size of 3x3 pixels. During the entire process, the DC-GAN generated a large quantity of images and their corresponding segmentation (151,040 generated couples), two examples are shown in Figure.1 b) and 1 c).

### 2.2.3. DATA AUGMENTATION PROTOCOL

We trained a SML method (see section 2.2.4) with 4 different augmentation protocols (Figure.2). First, the SML was trained with a dataset generated with DC-GAN that was trained with the learning dataset (protocol a). Then, the learning dataset was randomly split in two sets (protocols b, c and d). This split was controlled by a factor q, ranging from 0 to 1. The size of the first set was equal to the number of images in the learning dataset multiplied by q. An intermediate dataset, using this first set, was increased by a factor of 5 using a traditional method. For protocol b) SML was trained with the final learning dataset composed with the intermediate data set concatenated with the second set of the split. For protocols c) and d), GAN was trained with the intermediate dataset. In c) SML was trained with just a synthetic learning dataset coming directly from GAN. Finally, in d) SML was trained with a Hybrid Learning dataset composed with the synthetic learning dataset and the second set of the split of the learning dataset.

### 2.2.4. DATA AUGMENTATION VALIDATION

U-Net, a supervised deep learning segmentation method developed for biomedical image segmentation, was used to validate the data augmentation protocol (10). It was configured with default architecture parameters and trained with 35 epochs. A two-fold cross-validation (Direct Validation DV and Cross Validation CV) was performed through the computation of two F-scores (13). First, F-score Validation was computed during training, on a specific

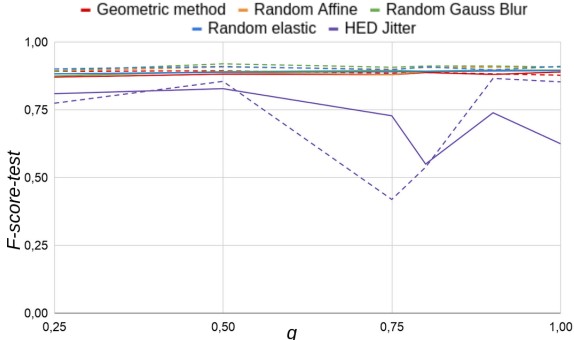

Figure 3: Final learning dataset (protocol b) F-score test (continuous line: DV. Dashed line: CV) against q values.

validation set (20 % of the final learning dataset) to detect potential overfitting (values close to 1). Finally, the F-score Test was evaluated using the test dataset, which was completely independent from the learning set. The model with the best F-score Validation during the training phase was used in the test phase.

## 3. Results

As a reference result, the F-Score test without augmentation was 0.877 in DV and 0.883 in CV.

### 3.1. DC-GAN based augmentation

The DC-GAN did not converge with the initial dataset (720 couple of images of 128x128 pixels) and generated unusable couples of images (Figure.2-c). In this case, F-score Validation was next to one and F-score Test was next to zero.

### 3.2. Traditional augmentation method

For protocol b, F-score Test plateaued above 0.889 in DV and 0.894 in CV for all the methods except HEDjitter (Figure.3). Moreover, the best F-score-test was achieved for q=1 which corresponded to the maximum amount of augmented images injected into the final learning dataset. Conversely, HEDjitter behavior was the opposite. The best F-score was generated with a dataset which contained less transformed images with this method (q=0.5).

The SSIM was used to sort traditional methods according to their power to structurally distort an image. The geometric, random affine and random elastic methods showed the lowest average SSIM value (Table 1).

| Traditional method | SSIM (DV) | SSIM (CV) |
|:---:|:---:|:---:|
| Geometric method | 0.44 ($\pm$0.05) | 0.46 ($\pm$0.05) |
| Random Affine | 0.44 ($\pm$0.07) | 0.45 ($\pm$0.08) |
| Random Gauss Blur | 0.93 ($\pm$0.04) | 0.92 ($\pm$0.04) |
| Random Elastic | 0.57 ($\pm$0.18) | 0.57 ($\pm$0.18) |
| HED Jitter | 0.81 ($\pm$0.09) | 0.82 ($\pm$0.09) |

Table 1: The average values of the metric SSIM calculated for each traditional method with the DV and CV mode, and the corresponding standard deviation.

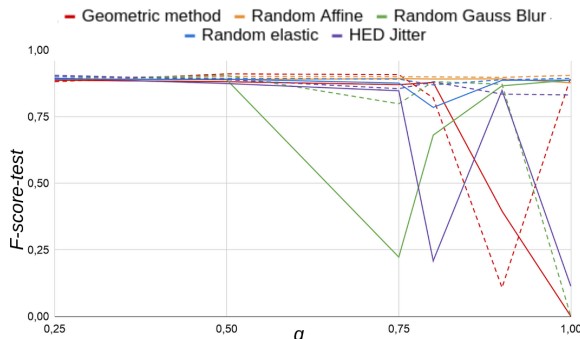

Figure 4: Hybrid learning dataset (protocol d) F-score test (DV continuous line and CV dashed line) against q values.

### 3.3. Hybrid and synthetic learning datasets

With a value of q lower than 0.25, the DC-GAN was not able to generate exploitable images. As for the hybrid learning dataset (Figure.4), the F-scores test in DV setting with Geometric method Random Gauss blur and HEDjitter method were not relevant because the DC-GAN could not converge. Conversely, the other methods presented high F-score values. The results of DC-GAN depended on the factor q, and generally increased with high q values. However, some DC-GAN failed to converge, even with q=1 (in CV setting, F-score tests were null for Random Gauss Blur and q=1). Furthermore, the Random Elastic method and the Random Affine method had a F-score higher than 0.89 in DV and 0.9 in CV for almost all q values.

Even when the DC-GAN generated only unusable images, U-net segmentation resulted in F-score test > 0, as data from the original learning dataset were injected in the learning process through the hybrid approach.

For the synthetic learning dataset using the Random Affine and Random Elastic method, F-score values stabilized asymptotically at approximately 0,87 in both DV and CV, following a horizontal plateau after a certain amount of data generated by the DC-GAN (1800 generated images of 128x128 pixels from 360 initial images, with q=0.5) (Figure.5). For the other methods, the stability of the convergence stage was not guaranteed.

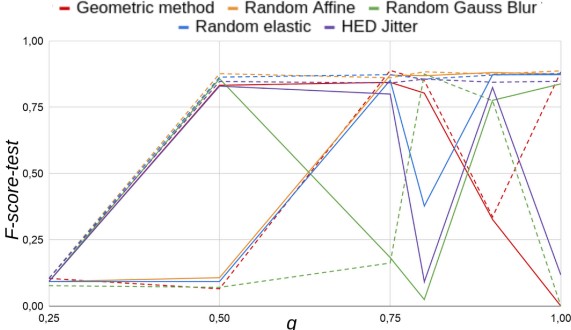

Figure 5: Synthetic learning dataset (protocol c) F-score test (DV continuous line and CV dashed line) against q values

## 4. Discussion

This study demonstrated the relevance of Data Augmentation strategies for SML algorithms in the context of histopathological images.

The geometric, random affine and random elastic methods led to low average SSIM values demonstrating an increase of spatial diversity in the subsequent generated-images. Contrarily to random gaussian blur and HEDJitter methods which changed only the color space and the blur of images, the three aforementioned methods affected the spatial distribution of the pixels. The geometric method obtained high results because SSIM was not rotation invariant (14). Therefore the spatial diversity measure with SSIM must not be considered equally between the geometric method and the couple formed by random affine and random elastic methods.

As shown in section 3.3, DC-GAN data augmentation approach allowed to reach similar segmentation quality than the reference result, but using only half of the initial dataset as an input. DC-GAN training results quality for BAM-10 datasets was linked with the quantity of data in the learning dataset. Without, or with a reduced traditional augmentation, a small initial learning dataset resulted in unusable images generated by the DC-GAN. While analyzing F-Scores (Figure.5), we noted that a minimum amount of data ( 1800 images of 128x128 pixels for q=0.5) was necessary to ensure exploitable results, as the GAN used an image by image strategy, and not pixel by pixel one. However, minimal input data quantity was a necessary condition but not sufficient condition to ensure the convergence of DC-GAN (Figure.5) and the generation of exploitable images.

When the random elastic method was used for learning dataset augmentation, DC-GAN convergence was ensured in almost all protocol cases, as already suggested elsewhere (15). Convergence not only depended on the quality of the original learning dataset, but also on the way augmentation methods extended datasets. Spatial deformation augmentation algorithms achieved high segmentation quality and asymptotic stability (Table 1, Figures 4 and 5). The worst algorithms for this purpose were HEDJitter and random gaussian blur, generating high-structural similarity images (Table 1).

Random elastic and affine methods increased learning dataset spatial diversity and therefore the quality of subsequent segmentation (15). Conversely, the other algorithms were not as stable and randomized as geometric methods, and only changed the colorimetric diversity (HED Jitter), but did not increase the geometric diversity (Random Gauss Blur).

## 5. Conclusion

In the light of these observations, we show that DC-GAN combined with traditional Data Augmentation methods is an efficient tool for histopathological dataset segmentation with a reduced initial manually labeled dataset. However the GAN learning dataset must meet two criteria (quantitative and qualitative) in order to obtain an efficient synthetic images generation. Firstly, a minimal quantity of initial learning images is necessary which corresponds to approximately 1,800 images of 128x128 pixels. Secondly, the spatial diversity of learning databases is mandatory to ensure high segmentation quality as shown by the SSIM sorting. Random affine and random elastic methods meet with this criterion.

Future work will consist in exploring other values of q in order to better specify the minimum quantity of data needed for GAN learning data, and reduce the minimal initial dataset size needed to ensure segmentation quality and stability. A particular attention will be given to the diversity of generated images, using the presented hybrid approach with the augmentation methods that work the best in collaboration with GANs. We will also test other GAN implementations, such as Progressive Growing GAN (PG-GAN) (16) and will evaluate inter GANs comparisons. Another lead will be to find data augmentation methods increasing further the spatial diversity and incorporate them in the presented pipeline. We will also investigate Transfer Learning between training GANs with different histological markers, allowing the generation of models for markers with the minimal amount of initial images.

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
