# OpenReview forum: "HISTOPATHOLOGY LEARNING DATASET AUGMENTATION: A HYBRID APPROACH"
_MIDL.io/2021/Conference — Submitted to MIDL 2021_

### Official Review · AnonReviewer2 · 2021-03-02

**Confidence:** 5
**Preliminary Rating:** 1
**Final Rating:** 1

**Summary:**

The paper presents a study of the affects of data augmentation via GAN based synthetic image generation and via traditional computer vision techniques. The authors present four distinct pipelines of using an existing dataset together with their augmented images to train a U-Net for histopathology segmentation.

**Strengths:**

The paper is correctly motivated, indeed getting vast quantities of annotated medical data is a non trivial task. The application the authors attempt to analyse is both an important one and a yet unsolved one.
The language of the paper is proper without any major issues.


**Weaknesses:**

There is no literature review, several works have been published through the last years on this topic, for example the 2017 work of Guibas et al. Synthetic Medical Images from Dual Generative Adversarial Networks. The authors are invited to first perform a thorough review of related works and incorporate them into their work by relating or comparing their approaches to prior work.

The presentation of the protocols the authors propose could have been clearer and better explained. It would have been quite helpful to itemize, compare and contrast the various approaches and most importantly explain what is the intuition and theoretical background of your choices.

Figs 3,4,5 are quite small to distinguish which line is on top of which.

The DC- GAN is a quite outdated approach with others like  Wasserstein-GAN solving mode collapse issues the DC-GAN was infamous for, what was the reasoning behind the choice of GAN method ?

The discussion relating to the results is somewhat superficial, what is the take home message from fig 3,4,5? Is there an explanation on why we observe these dips in performance in high q values ?

There is mention about the qualitative evaluation of the resulting datasets, but no visual examples are produced in the paper. How have the authors made sure that the resulting images they produce are reasonable ?

**Deanonymize Review:**

no

**Final Rating Justification:**

No rebuttal was introduced, I maintain my original review and rating

**Justification Of The Preliminary Rating:**

Overall this work seems to have been done in a rush. Major elements are missing and comparison with other advanced data augmentation methods is not included. The visual quality or the rationality of the produced dataset is not explored. It is a good first step but more work is needed to improve this paper. As the major claim is that the paper conducts a study of data augmentation using synthetically generated datasets , it would have been ideal to have some intuition on why the proposed methods work as they should and what are their limitations.

**Paper Type:**

validation/application paper

**Special Issue:**

no

---

### Official Review · AnonReviewer4 · 2021-03-06

**Confidence:** 5
**Preliminary Rating:** 2
**Recommendation:** Poster

**Summary:**

The paper proposes a data augmentation method for histopathology images that is a hybrid approach of transformation-based data augmentation and generative adversarial networks. The topic that the paper addresses is very relevant in histopathology image analysis, as indeed there are some problems/applications where only limited dataset sizes are available. That being said, I am not completely convinced in the utility of thee proposed method.

**Strengths:**

The paper tackles a very relevant problem in histopathology image analysis. The authors have performed a relatively extensive set of experiments. The proposed methodology (combination of classical data augmentation and GANs) is very straightforward and logical.

**Weaknesses:**

- The paper is not very clearly written and is not easy to follow.
- The segmentation problem (essentially segmentation of positive stain) that is used to evaluate the proposed augmentation method is not very challenging. A more challenging problem would have been more convincing.

**Deanonymize Review:**

no

**Detailed Comments:**

- Some components of the paper are very unclear. What is the intermediate dataset?
- Figure 1 b and c are not referenced in the text. Figure 1 c seems odd and like the result from a GAN that was not trained properly.
- It is unclear if the data was split at the subject level or at the patch level. The latter is the proper way to go, the former will lead to optimistic estimates of the performance.

**Justification Of The Preliminary Rating:**

There might be some merit to the propose work, however, the presentation is very unclear. Given the difficulty of the problem to which this methodology was applied I am not very convinced in the utility.

**Paper Type:**

methodological development

**Questions To Address In The Rebuttal:**

It would be good to see this applied to a more challenging problem. The unclarity about the separate of the data into subsets (see above) must be resolved. The authors should explain the methodology in a more concise and clear way.

**Special Issue:**

no

---

### Official Review · AnonReviewer3 · 2021-03-09

**Confidence:** 4
**Preliminary Rating:** 1
**Final Rating:** 1

**Summary:**

In this submission, the authors evaluate a data augmentation strategy based on DC-GANs along with a number of not-learning-based augmentation methods for stain quantification in histopathology images. Different combinations of augmentation strategies were investigated as input for a subsequent U-Net training. Different strategies for combining the augmentation techniques were evaluated. The authors conclude that DC-GAN in combination with non-learning augmentation techniques are en efficient tool for histopathological dataset augmentation.

**Strengths:**

+ Interesting overview of available augmentation methods in the introduction
+ Good comparison of different augmentation techniques for this problem
+ Interesting observation that HEDjitter behaves differently from the other techniques.

**Weaknesses:**

- I am not able to discern the methodological novelty with respect to very related work (that the authors were perhaps not aware of (see detailed comments below)).
- DC-GAN (Radford et al., 2015) while an important contribution at the time, cannot be considered state-of-the-art anymore. Thus insights obtained using this technique may be limited.
- Certain aspects of the method are not described in sufficient detail.


**Deanonymize Review:**

no

**Detailed Comments:**

GAN based image augmentation has become a whole research field onto itself and this paper, fails to acknowledge and draw from the extensive amount of work in this field that has already been performed.
To name a few examples:

 - Frid-adar et al., 2018 https://www.sciencedirect.com/science/article/abs/pii/S0925231218310749
 - Chaitanya et al., 2019 https://link.springer.com/chapter/10.1007/978-3-030-20351-1_3
 - Mahmood et al., 2018 https://ieeexplore.ieee.org/abstract/document/8370747

And here a few examples specifically focused on histopathology segmentation:

 - Hou et al., 2017 https://arxiv.org/abs/1712.05021
 - Xue et al., 2021 https://www.sciencedirect.com/science/article/abs/pii/S1361841520301808
 - Senaras et al., 2018 https://journals.plos.org/plosone/article?id=10.1371/journal.pone.0196846

I urge the authors to acknowledge this work and consider how their research fits into this landscape.

Related to the point above, I believe DC-GAN cannot be considered state-of-the art anymore given the vast amount of work in the domain of GANs. Thus results obtained with DC-GAN are not very informative because a modern GAN approach may behave very differently (perhaps more favourably)

I am not sure what insights the authors expect to obtain from comparing SSIM between the different augmentation techniques? As the authors mention themselves SSIM is a very limited measure and already behaves unexpectedly in simply cases such as rotation. It is stated that low SSIM means higher spatial diversity. But it be concluded that this will be better for learning? That is, did the authors find SSIM to be predictive of usefulness for the subsequent supervised learning step?

The text states that "generation started after the 360th epochs and 100 images per epoch, until the generation of the final dataset." I am confused by this statement. Does that mean that images were generated while the training was still ongoing?

Please state in more detail how the GAN training was set up? The GAN generates images and corresponding segmentation masks. How were the two generated together (concatenated in channel direction?), and how did the authors ensure that the synthetic segmentation mask fit the synthetic image?

The authors mention that they were limited by the amount of data and in some instances were not able to train the GAN due to lack of data. Could the authors not simply have extracted more patches from the digitised sections? To my understanding, this was an arbitrary number of random patches, correct?

I found the data augmentation protocol (Sec 2.2.3.) extremely difficult to follow. Perhaps, a short intuitive explanation of what the authors aim to achieve with the different splits, would help to follow the technical steps.


**Final Rating Justification:**

This paper has to my understanding been withdrawn. In any case no rebuttal --> no change in score.

**Justification Of The Preliminary Rating:**

I don't believe this work is ready for publication at MIDL. While the paper certainly solves an important problem, the work should be reconsidered in the light of many recent very related papers. I could imagine that a similar analysis using one of the more recent GAN based histopathology augmentation frameworks would be of interest to the community.

**Paper Type:**

both

**Special Issue:**

no

---

### Meta-Review · Area_Chair1 · 2021-03-24

**Recommendation:** Reject

**Metareview:**

The reviewers are in agreement that the paper should be rejected. Furthermore, the authors have declined to rebut the arguments of the reviewers. As such, I agree with the suggested rejection.

**Paper Type:**

validation/application paper

---

### Decision · Program_Chairs · 2021-03-31

Reject